# Molecular Modeling of Pathogenic Mutations in the Keratin 1B Domain

**DOI:** 10.3390/ijms21186641

**Published:** 2020-09-10

**Authors:** Alexander J. Hinbest, Sherif A. Eldirany, Minh Ho, Christopher G. Bunick

**Affiliations:** 1Department of Molecular Biology and Biochemistry, Wesleyan University, Middletown, CT 06459, USA; ahinbest@wesleyan.edu; 2Yale School of Medicine, Yale University, New Haven, CT 06520, USA; sherif.eldirany@yale.edu; 3Department of Dermatology, Yale University, New Haven, CT 06520, USA; minh.ho@yale.edu; 4Department of Molecular Biophysics and Biochemistry, Yale University, New Haven, CT 06520, USA

**Keywords:** keratin, intermediate filament, structure, modeling, skin disease, liver disease, epidermolysis bullosa, cytoskeleton, mutation

## Abstract

Keratin intermediate filaments constitute the primary cytoskeletal component of epithelial cells. Numerous human disease phenotypes related to keratin mutation remain mechanistically elusive. Our recent crystal structures of the helix 1B heterotetramer from keratin 1/10 enabled further investigation of the effect of pathologic 1B domain mutations on keratin structure. We used our highest resolution keratin 1B structure as a template for homology-modeling the 1B heterotetramers of keratin 5/14 (associated with blistering skin disorders), keratin 8/18 (associated with liver disease), and keratin 74/28 (associated with hair disorder). Each structure was examined for the molecular alterations caused by incorporating pathogenic 1B keratin mutations. Structural modeling indicated keratin 1B mutations can harm the heterodimer interface (R265P^K5^, L311R^K5^, R211P^K14^, I150V^K18^), the tetramer interface (F231L^K1^, F274S^K74^), or higher-order interactions needed for mature filament formation (S233L^K1^, L311R^K5^, Q169E^K8^, H128L^K18^). The biochemical changes included altered hydrophobic and electrostatic interactions, and altered surface charge, hydrophobicity or contour. Together, these findings advance the genotype-structurotype-phenotype correlation for keratin-based human diseases.

## 1. Introduction

Intermediate filaments (IFs) work with actin microfilaments and microtubules to provide essential cytoskeleton functions within eukaryotic cells. IFs are aptly named because their diameter (~10 nm) is in between that of actin (~4–5 nm) and microtubules (~25 nm) [1]. There are six types of IFs [2], with type I and II representing keratins. The 54 keratin genes outnumber all the other IF types combined. Keratins are obligate heterodimers, with one type I keratin dimerizing in register and parallel with one type II keratin [3]. Parallel heterodimers then form anti-parallel tetramers, which serve as the building block for mature IFs.

Keratins are differentially expressed across human tissues, providing unique biological functions to support the host organism. For example, keratins 5 and 14 (K5/14) and K1/10 are expressed in stratified epithelia. K5/14 are the primary keratins in basal layer keratinocytes, whereas K1/10 are the primary suprabasal IF proteins in differentiating keratinocytes of the epidermis. Hard keratins (K25–86), found in hair and nails, utilize abundant cysteine cross-linking to form more rigid filaments than epithelial keratin IFs (K1–K24) [4].

Keratins support human health, as exemplified by the fact that more than 80 human IF-related diseases (termed IF-pathies [5]) exist. For example, different keratin mutations are responsible for blistering skin disorders (e.g., epidermolysis bullosa simplex, EBS), keratodermas (e.g., epidermolytic palmoplantar keratoderma, EPPK), hair and nail defects (e.g., pachyonychia congenita), and liver disease (e.g., cryptogenic cirrhosis) [6]. Keratins have also been linked to cancer, corneal dystrophy, and pancreatitis [7]. IF mutations vary in type: deletion, insertion, nonsense, and missense. Efforts have been made to correlate a patient’s keratin genotype with clinical phenotype for these diseases [7,8,9,10,11,12,13]. Understanding how keratin genotype impacts the molecular structure of keratin filaments is also important, and we have termed this concept genotype-structurotype-phenotype correlation [14,15].

Keratins share a common protein organization: flexible, glycine- and serine-rich head and tail domains bookend a highly-conserved central coiled-coil rod domain, itself comprised of 1A, 1B, 2A, and 2B subdomains. We recently determined two keratin 1/10 1B domain crystal structures [16]. The first was a 3.0 Å resolution structure of wild-type K1/K10-1B (Protein Data Bank (PDB) ID 6EC0), and the second was a 2.4 Å resolution structure of K1/K10-1B incorporating the S233L^K1^ mutation responsible for EPPK (PDB ID 6E2J). 1B domain structures of homomeric IF proteins have also been determined (vimentin, glial fibrillary acidic protein (GFAP), lamin A) [17,18,19,20,21]. The keratin structures represent the only heteromeric 1B structures to date, and as such they are experimentally determined quality templates for modeling 1B domains from other keratin pairs. Here, we used our highest resolution K1/K10 1B structure to model 1B domains of K5/K14, K8/K18, and K74/K28 in order to evaluate the structural alterations associated with pathogenic keratin missense mutations in this region. In particular, we wanted to ascertain whether the mutations would impact the dimer, tetramer, or high-order assembly of the keratins, and how the mutations would affect the molecular surface properties (charge, hydrophobicity, contour) of the keratins.

## 2. Results

### 2.1. Identification of Keratin 1B Mutations

The Human Intermediate Filament Database (HIFD) [6] was analyzed to find all missense mutations in the 1B region of keratins. We identified 15 mutations: 9 missense, 3 nonsense, 2 leading to frameshift, and 1 in-frame deletion (indel) (Table 1). To predict the potential impact of the missense mutations on keratin structure and function, we utilized Polyphen-2 to analyze the primary keratin sequences [22]. Six out of nine keratin missense mutations had a polyphen score classified as possibly or probably damaging (F231L^K1^, S233L^K1^, R265P^K5^, L311R^K5^, R211P^K14^, F274S^K74^) (Table 1). The other three mutations (Q169E^K8^, H128L^K18^, I150V^K18^) were predicted to be benign.

### 2.2. K1/K10 1B Missense Mutations Associated with Keratodermas

We previously determined the crystal structure of K1/K10 containing S233L^K1^ and also characterized the 1B domain harboring S233L^K1^ by multi-angle light scattering (MALS) [16]. This particular mutation is associated with epidermolytic palmoplantar keratoderma (EPPK) and histologically has the key finding of tonotubular keratin [24,42]. The structure identified that S233L^K1^ caused a new hydrophobic patch on the molecular surface of the K1/K10 dimer, and MALS demonstrated it also caused keratin aggregation in solution. The structure also showed that the aberrant leucine residue directly participated in inter-tetramer contacts promoting octameric assembly between K1/K10 tetramers (Figure 1a). How this excess hydrophobicity on the K1/K10 dimer surface leads to tubule formation for full-length K1/K10 remains unclear, but the tendency for hydrophobicity-driven aggregation is established.

In contrast to S233L^K1^, F231L^K1^ alters the tetrameric interface between two K1/K10 dimers at the anchoring knob-hydrophobic pocket assembly mechanism [16]. F231L^K1^ is associated with non-epidermolytic palmoplantar keratoderma (NEPPK) [23]. In the wild-type K1/K10 complex, F231^K1^ makes molecular contacts with F314 and L318 from the partner K1 molecule to help form the K1/K10 tetramer (Figure 1b). Modeling of F231L^K1^ suggests there is an increase in the contact distance between the mutant L231 and L318’ as well as a loss of the aromatic–aromatic interactions by F314’ (Figure 1c). These molecular changes caused by F231L^K1^ likely destabilize the keratin 1B tetrameric interface, which in turn affects mature filament formation and function.

### 2.3. K5/K14 1B Missense Mutations Cause Epidermolysis Bullosa Simplex

Mutations in keratins 5 and 14 are commonly associated with EBS, but the vast majority of these mutations lie within the 1A or 2B subdomains [13]. Of the 9 unique mutations we identified in the 1B subdomain (Table 1), three were in K5/K14: two K5 (R265P, L311R) and one K14 (R211P) missense mutations that are documented to cause EBS (Figure 2).

R211P^K14^ was identified during screening of 27 German EBS patients; the individual from which the mutation was identified also suffered from palmoplantar symptoms [35]. The authors hypothesized that proline incorporation due to R211P^K14^ led to helix kinking, which is a known structural property of proline [43]. R211^K14^ lies in the center of the K5/14 1B domain at the heterodimer interface (Figure 2a). Modeling of wild-type K5/14 1B shows R211 forms electrostatic interactions or salt bridges with E215^K14^ and E261^K5^ (Figure 2c). By having intra-K14 and inter-K5 interactions, R211^K14^ provides stabilization to the K5/14 1B dimer interface. Mutation to proline eliminates the dual electrostatic/salt bridge interactions, in addition to kinking the K14 helix, thereby destabilizing the K5/14 heterodimer.

R265P^K5^ is a mutation identified in Korean EBS patients [26] and located in the middle of the 1B domain (Figure 2a). The structural model of K5/14 1B demonstrates that R265^K5^, like R211^K14^, forms electrostatic interactions or salt bridges with residues that are intra-strand (E269^K5^) as well as opposite dimer strand (E207^K14^) (Figure 2d). Mutation R265P^K5^ eliminates these important dual interactions that stabilize the K5/14 heterodimer, in addition to the expected helix kinking.

L311R^K5^ was one of several K5/14 mutations detected from sequencing of 10 Israeli patients suffering from EBS [27]. L311^K5^ is located at the C-terminus of the K5/14 1B heterodimer, where it makes hydrophobic interactions in the dimer interface with K14 residues H253, M257, and L260 (Figure 2b). L311R^K5^ mutation leads to the substitution of a bulkier basic residue that disrupts hydrophobic interactions and may generate new electrostatic interactions with E256^K14^. The molecular surface properties of K5/14 1B are also changed, notably the loss of a surface-exposed hydrophobic patch at the C-terminus of the K5/14 1B heterodimer (Figure 2e). The loss of surface hydrophobicity likely disrupts higher-order filament interactions needed for mature IF formation.

### 2.4. Structural Alterations in K8/K18 1B Are Associated with Liver Disease

Keratins 8 and 18 (K8/18) are predominantly expressed in “simple” epithelia, and mutations in them are frequently associated with liver disease (Table 1 and Figure 3) [44]. Q169E^K8^ was identified during sequencing of 162 hemochromatosis patients as part of an analysis of K8/18 intron mutations associated with hereditary hemochromatosis or liver fibrosis [28]. Q169^K8^ is located in the central aspect of the K8/18 1B domain as part of the heterodimer interface (Figure 3a). It makes local van der Waals interactions with L165^K8^ and Q153^K18^ (Figure 3c). Q169E^K8^ does not appear to significantly alter the local heterodimer packing, although the OE2 atom on Q169E^K8^ might form a hydrogen bond with the Q153^K18^ NE2 atom. The larger effect of Q169E^K8^ is on the molecular surface, where the acidic nature of the K8/K18 1B dimer and tetramer is enhanced by the mutation (Figure 3e). This acidic patch created by Q169E^K8^ may disturb higher-order K8/K18 filament assembly.

H128^K18^ is located at the N-terminus of the K8/18 1B heterodimer (Figure 3a). It is solvent-exposed and does not contribute to the hydrophobic dimeric interface (Figure 3b). It also faces solvent in the K8/18 1B tetramer, suggesting that H128L^K18^ mutation does not destabilize the K8/18 dimer or tetramer, but rather alters the higher-order assembly of mature filaments. H128L^K18^ correlated with liver cirrhosis in a study involving patients with K8 and K18 mutations [45]. Mapping of hydrophobic potential onto the molecular surface of the K8/18 1B tetramer demonstrated that H128L^K18^ creates a bulky surface-exposed hydrophobic patch (Figure 3f). This patch would occur at both ends of the K8/18 1B tetramer due to the antiparallel alignment between dimers.

I150V^K18^ is another K18 mutation associated with liver disease [40]. I150^K18^ is located in the center of K8/18 1B at the heterodimer interface (Figure 3a). It makes several hydrophobic contacts to stabilize the dimer, including interactions with I154^K18^, L161^K8^, and L165^K8^. The CD1 atom of I150^K18^ also interacts with the aliphatic portions of K158^K8^ and E162^K8^, which stabilizes K158^K8^ so that it can make a salt bridge with D146^K18^ (Figure 3d). I150V^K18^ preserves the core hydrophobic interactions with I154^K18^, L161^K8^, and L165^K8^, but loss of the CD1 atom likely destabilizes contacts with K158^K8^ and E162^K8^ and potentially compromises the E162^K8^-D146^K18^ salt bridge.

### 2.5. Anchoring Knob Mutation in the Hard Keratin K74

Keratin 74 (K74) is a type II IF protein whose obligate IF partner has yet to be unambiguously identified. Prior work showed that K74 can pair with K14 and K18 [46]. However, this is inconsistent with the fact that K74 is largely expressed in the inner root sheath (IRS) of hair, while K14 and K18 are expressed in basal keratinocytes and simple epithelia, respectively. There is evidence for K74 co-expression with K28 [47], which more closely aligns with the physiology of hair keratin IF formation (both are inner root sheath keratins). Therefore, we modeled the K74/28 1B heterotetramer (Figure 4).

F274S^K74^ was identified during sequencing of a consanguineous Pakistani family which had homozygous autosomal recessive pure hair and nail ectodermal dysplasia (PHNED) [41]. Patients with PHNED displayed loss of K74 when staining for expression in nail matrix, nail bed, and IRS of hair follicle, in addition to mouse hyponychium. With the available information at the time, it was hypothesized that F274S^K74^ would interfere with keratin hetero-dimerization. However, our recent crystal structures of K1/K10 1B tetramers (wild-type and S233L^K1^) demonstrated that the sequence position of F274^K74^ forms part of the anchoring knob involved in the anchoring knob–hydrophobic pocket tetramer assembly mechanism (Figure 4a,b). This mechanism is important for A_11_ tetramer formation (where A_11_ means the 1B domains are antiparallel and aligned in-phase in the tetramer) and mature filament assembly [16].

In the structural model, K74 residues F274’ and L278’ form the anchoring knob at the K74/28 tetramer interface (Figure 4c). These residues pack into a hydrophobic pocket formed by the partner heterodimer (K74 residues L187, I190, L191, and Y194, and K28 residue I144). F274S^K74^ mutation dramatically alters the knob-pocket interaction site, eliminating hydrophobic interactions and aromatic–aromatic interactions (Figure 4d). This will destabilize the K74/28 1B tetramer at both ends. Thus, F274S^K74^ is likely to severely alter mature filament assembly based on prior electron microscopy data showing keratins 1/10 and 8/18 and vimentin 1B knob mutations damage IF assembly [16].

## 3. Discussion

With 54 genes and differential expression by time and tissue type, keratins play a significant, but complex role in human physiology. Their medical relevance is exemplified by the high numbers of IF-related diseases associated with keratin mutation [5,7]. Their role in establishing and maintaining skin, hair, and nail aesthetics is also important. How keratins promote structural integrity within cells, facilitate protein interactions and signaling, and regulate tissue properties remains poorly understood at a molecular level. Their long, filamentous and insoluble nature contributes greatly to the difficulty in studying these proteins. Despite all the keratins known, only four out of 54 (K1, K10, K5, K14) have experimentally determined atomic resolution structures [12,13,14,16,48]. In total, there are six crystal structures of human keratins. Only one of these contains a mutation (S233L^K1^) known to cause human disease [16]. Therefore, a better understanding of how keratin mutations alter protein structure is needed to help correlate patient genotype with phenotype: a paradigm we call genotype-structurotype-phenotype correlation [14,15].

In the spirit of this paradigm, the work here examines nine pathologic keratin mutations that occur within the 1B subdomain of the keratin coiled-coil rod domain. Most keratin mutations associated with clinical disease are located in the 1A and 2B subdomains, with only a small number to date occurring in 1B. The molecular explanation for this difference is not clear, but evidence suggests the 1B domain is critical for establishing the tetramer building block of IFs, whereas the 1A and 2B domains may play a greater role in the higher-order packing interactions needed to form mature IFs [14,16,48,49]. In our analysis of the keratin 1B mutants, we considered: (1) What interface do the mutations affect, and (2) what do the mutations do to the molecular surface chemistry of the keratin?

Structural modeling indicates that some keratin 1B mutations harm the heterodimer interface (R265P^K5^, L311R^K5^, R211P^K14^, I150V^K18^), some harm the tetramer interface (F231L^K1^, F274S^K74^), and some harm higher-order interactions needed for mature IF formation (S233L^K1^, L311R^K5^, Q169E^K8^, H128L^K18^). Some mutations, like L311R^K5^, alter multiple interfaces. Three of the 1B mutations have a prominent change in molecular surface properties: Q169E^K8^ generates an acidic surface patch, L311R^K5^ eliminates a hydrophobic patch, and H128L^K18^ creates a new hydrophobic patch. Solvent accessible hydrophobic surface has been shown to cause aberrant aggregation for S233L^K1^ [16], leading to dramatic morphological change from tonofilaments to tonotubules [24]. H128L^K18^ altered K8/K18 IFs in vitro by electron microscopy [39], likely due to aberrant hydrophobic interactions. Importantly, our analysis demonstrated structural changes conducive to IF pathology even for the three missense mutations (Q169E^K8^, H128L^K18^, I150V^K18^) allocated a “benign” prediction by Polyphen-2 analysis (Table 1). This indicates that primary sequence analysis alone is not sufficient to capture the pathogenicity of a keratin mutation, and that analysis of keratin structure plays an essential role in understanding the molecular basis of disease.

The molecular modeling presented here establishes a foundation for designing future biochemical studies to validate the predicted amino acid properties of these keratin mutations. While the modeling is in silico, this computational structural biology is based on an experimentally derived keratin 1/10 tetrameric crystal structure, and thus is more reliable than ab initio structure prediction. The analysis of pathogenic K1 mutations mapped onto the 1B structure advances our knowledge of how disease-causing keratin mutations alter the various stages of mature filament assembly.

Ultimately, each pathologic keratin mutation will exhibit its own “structurotype” or set of biochemical and structural changes driving pathogenesis. Identifying the molecular parameters of a keratin mutation is the initial step toward the goal of being able to manipulate keratin IFs for pharmacological purposes [50]. As the number of experimental keratin structures grows, so will the insights into the molecular mechanisms driving mature filament assembly. This knowledge is essential for understanding IF-related diseases and developing new approaches to treat keratinopathies.

## 4. Materials and Methods

Keratin primary sequences, mutations, and corresponding literature were identified using the HIFD (interfil.org) [6] and cross-referenced using the UniProt [51] and National Center for Biotechnology Information (NCBI) variant databases: dbSNP [52] and ClinVar [53]. NCBI Protein Accession Numbers for each keratin analyzed were: K1: NP_006112.3, K10: NP_000412.3, K5: NP_000415.2, K14: NP_000517.2, K8: NP_002264.1, K18: NP_000215.1, K74: NP_778223.2, K28: NP_853513.2. Polyphen-2 was used to assess mutation severity [22]. Clustal Omega was used to perform multiple sequence alignments [54]. Homology modeling was performed with SWISS-MODEL [55] using the K1(S233L)/K10-1B tetramer structure (PDB ID 6E2J) as a template [16]. Coot [56] and UCSF Chimera [57] were used for energy minimization, structure analysis, and figure preparation. Final figures were made using Adobe Illustrator.

## Figures and Tables

**Figure 1 ijms-21-06641-f001:**
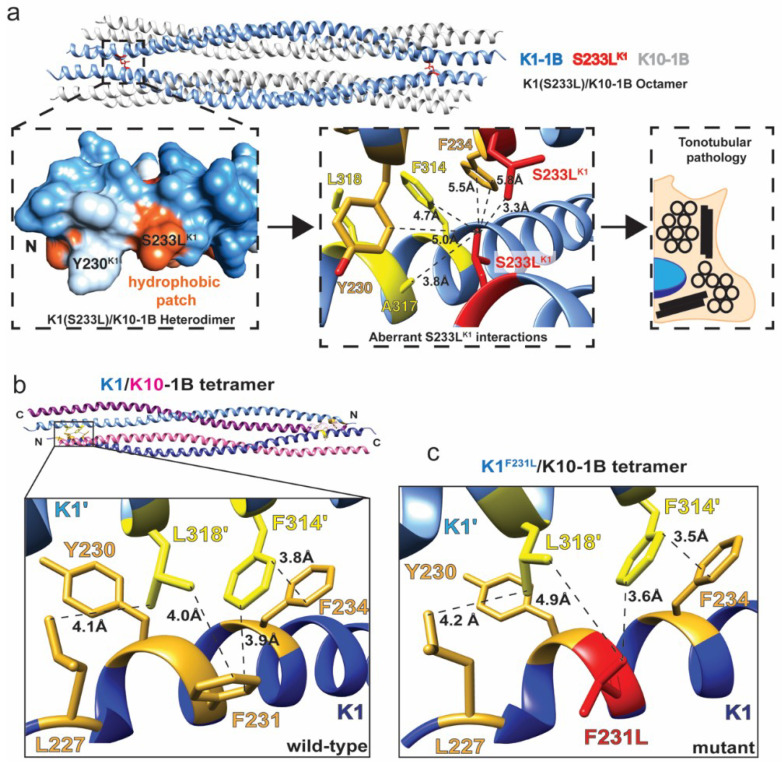
Keratin 1 1B mutations affect the oligomeric state of the K1/K10-1B heterocomplex. (**a**) The crystal structure of the K1^S233L^/K10-1B domain (PDB ID 6E2J) is an octamer composed of two tetramers (top). A surface-exposed hydrophobic patch is created by the serine to leucine mutation (left), resulting in the leucine making multiple aberrant hydrophobic interactions between tetramers (center). Histologically, this mutation results in tonotubular keratin, a more aggregated form than wild-type tonofilaments (right). (**b**) The crystal structure of the wild-type K1/K10-1B domain (PDB ID 6E2J) is a tetramer composed of anti-parallel dimers. A key interaction occurs between heterodimers: K1 residues comprised of an anchoring knob from one dimer (F314, L318, yellow) bind into a hydrophobic pocket on the surface of the other dimer (L227, Y230, F231, F234, gold). (**c**) The F231L mutation occurs in one of the key hydrophobic pocket residues in K1 (red). Modeling of this mutation reveals disruption of the knob-pocket interaction, likely leading to tetramer destabilization.

**Figure 2 ijms-21-06641-f002:**
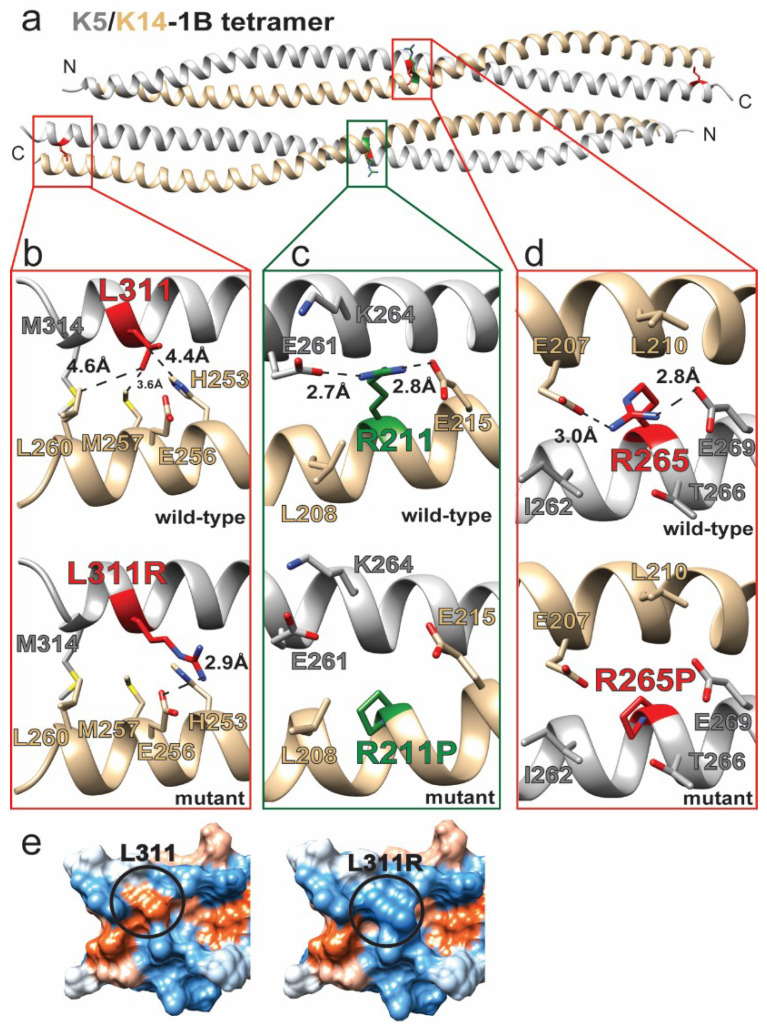
Keratin 5 and keratin 14 1B mutations destabilize the K5/K14 heterodimer. (**a**) The K5/K14-1B tetramer was generated by homology modeling using the crystal structure of K1/K10-1B (PDB ID 6E2J) as a template. Two of the three identified missense mutations occur in the middle of the domain and the other at the carboxy-terminal end. (**b**) The L311R^K5^ mutation disrupts intra-dimeric hydrophobic interactions with nearby K14 residues and may cause an aberrant electrostatic interaction (with E256^K14^). (**c**) R211P^K14^ disrupts two electrostatic interactions (with E261^K5^ and E215^K14^) at the dimer interface. (**d**) R265P^K5^ disrupts two electrostatic interactions (with E269^K5^ and E207^K14^) at the dimer interface. (**e**) The comparison of hydrophobic surfaces (orange = hydrophobic, white = neutral, blue = polar) of the wild-type (left) and L311R^K5^ (right) K5/K14-1B C-terminus demonstrates that the mutation eliminates a surface-exposed hydrophobic patch.

**Figure 3 ijms-21-06641-f003:**
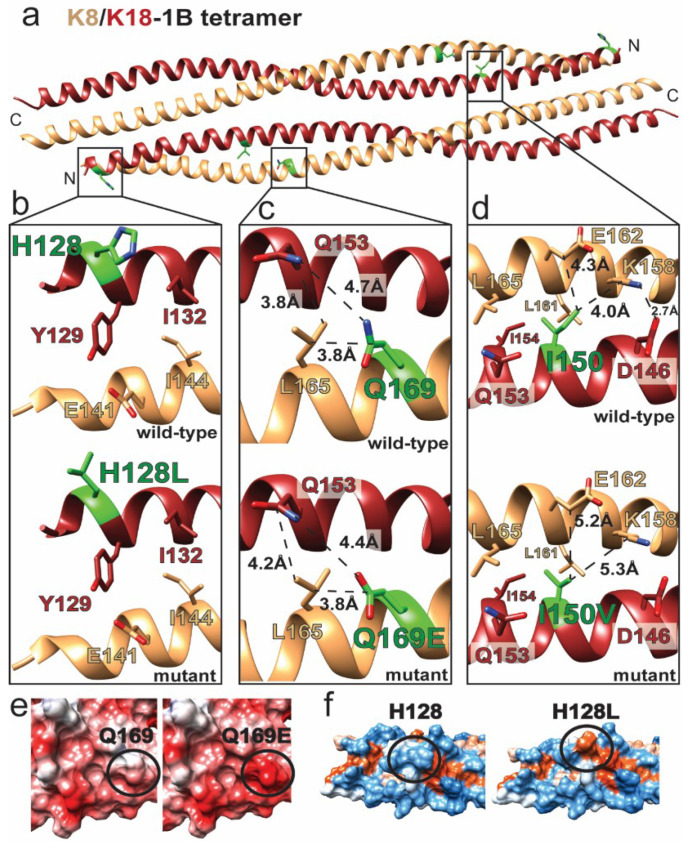
Keratin 8 and keratin 18 mutations alter intra-dimeric interactions and molecular surface properties. (**a**) The K8/K18-1B tetramer was generated by homology modeling using K1/K10-1B (PDB ID 6E2J) as a template. Missense mutated residues are identified in green. (**b**) The H128L^K18^ mutation does not disrupt intra-dimeric or tetrameric interactions. (**c**) Q169E^K8^ has limited impact on packing at the dimer interface. (**d**) I150V^K18^ preserves most hydrophobic interactions at the dimer interface but may destabilize a salt bridge at the dimer interface (between K158^K8^ and D146^K18^) by eliminating a contact with K158^K8^. (**e**) Comparison of electrostatic surfaces (red = acidic, white = neutral, blue = basic) of the wild-type (left) and Q169E^K8^ (right) K8/K18-1B tetramer demonstrates that the mutation enhances the surface acidity of the molecule. (**f**) The comparison of the hydrophobic surfaces (orange = hydrophobic, white = neutral, blue = polar) of the wild-type (left) and H128L^K18^ (right) K8/K18-1B tetramer demonstrates that the mutation introduces a surface-exposed hydrophobic patch.

**Figure 4 ijms-21-06641-f004:**
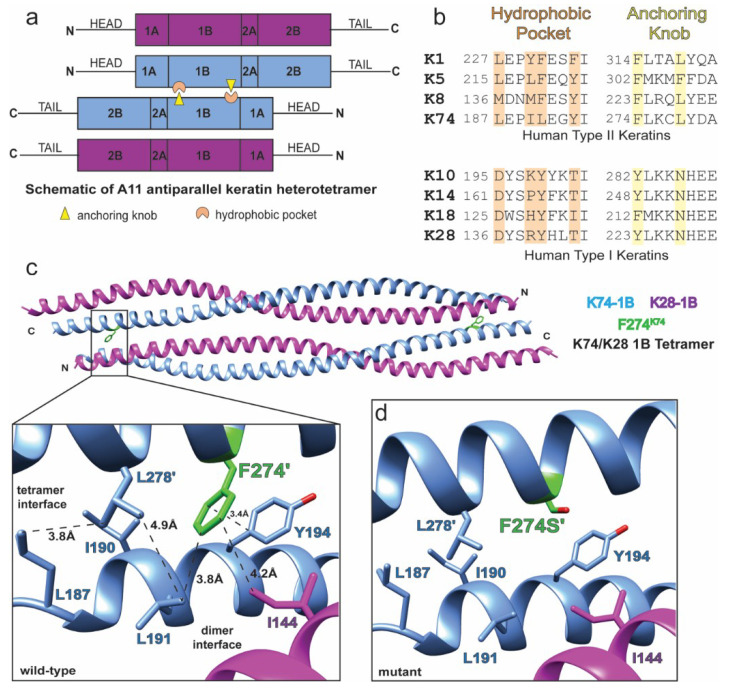
The F274S^K74^ mutation occurs in the anchoring knob and disrupts the tetramer interface. (**a**) The antiparallel keratin heterotetramer is stabilized by an anchoring knob-hydrophobic pocket interaction at both ends of the 1B domain. The “A11” alignment refers to the 1B domain of each heterodimer being in phase with each other in the tetramer state. (**b**) Knob and pocket residues are largely conserved in the type II keratins. These residues are not conserved in the type I keratins. (**c**,**d**) Modeling of the K74/K28-1B tetramer demonstrates loss of hydrophobic interactions in the knob-pocket mechanism as a result of the F274S^K74^ mutation.

**Table 1 ijms-21-06641-t001:** Keratin 1B domain mutations identified from the Human Intermediate Filament Database. Fifteen DNA and protein mutations are listed, with 9 missense mutations that were modeled in this study. The Polyphen-2 algorithm was used to generally predict a mutation effect on the keratin. “Disease Mechanism” depicts the hypothesized or experimentally validated filamentous structural changes believed to contribute to the disease phenotype. Abbreviations: (N)EPPK, (Non)-epidermal Palmoplantar Keratoderma; BCIE, Bullous Congenital Ichthyosiform Erythroderma; EHK, Epidermolytic Hyperkeratosis; (R)EBS-K/WC, (Recessive) Epidermolysis Bullosa Simplex: Koebner or Weber–Cockayne subtypes; CIEH, Cyclic Ichthyosis with Epidermolytic Hyperkeratosis; ADWH, Autosomal-Dominant Woolly Hair; LOF, Loss of Function.

DNA Mutation	Protein Mutation	Protein	Hetero Pair	Mutation Type	PolyphnScore	PolyphenPrediction	Disease Association	Disease Mechanism	References
c.693T>G	p.Phe231Leu	K1	K1/10	Missense	0.971	Probably Damaging	NEPPK	Disrupts 1B knob-pocket interaction	[23]
c.698C>T	p.Ser233Leu	K1	K1/10	Missense	0.704	Possibly Damaging	EPPK, NEPPK, BCIE/EHK	Tonotubular filament formation	[16,23,24,25]
c.794G>C	p.Arg265Pro	K5	K5/14	Missense	0.975	Probably Damaging	EBS-K	Unknown	[26]
c.932T>G	p.Leu311Arg	K5	K5/14	Missense	0.908	Possibly Damaging	EBS-WC	Unknown	[27]
c.475C>G	p.Gln169Glu	K8	K8/18	Missense	0.171	Benign	Cryptogenic cirrhosis	Unknown	[28]
c.846T>A	p.Tyr282X	K10	K1/10	Nonsense (truncation)	N/A	N/A	CIEH	LOF (K10 K/O), aggregated K1, compensatory upregulation of K14 & K17	[29]
c.526-2A>C	p.[Ile176ValfsX2, Ile176ProfsX30]	K14	K5/14	Frame-shift (truncation)	N/A	N/A	REBS, REBS-K	Basal K14 IF loss, compensatory K15 protofilaments	[30,31,32]
c.[612T>A]+[612T>A]	p.[Tyr204X]+[Tyr204X]	K14	K5/14	Nonsense (truncation)	N/A	N/A	REBS-K	"Natural K14 K/O", Basal K14 IF loss, insoluble keratin aggregation	[33,34]
c.632G>C	p.Arg211Pro	K14	K5/14	Missense	0.999	Probably Damaging	EBS-WC	Unknown	[35]
c.740_748delCCTACCTGAinsGAA	p.Ala247_Lys250delinsGlu	K14	K5/14	Indel (in-frame)	N/A	N/A	EBS-WC	Unknown	[36]
c.744delCinsAG	p.Tyr248X	K14	K5/14	Nonsense (truncation)	N/A	N/A	REBS-K	"Natural K14 K/O", Basal K14 IF loss, insoluble keratin aggregation	[37]
c.749delA	p.Lys250ArgfsX8	K14	K5/14	Deletion (frame-shift)	N/A	N/A	REBS	Absent K14 expression	[38]
c.383A>T	p.His128Leu	K18	K8/18	Missense	0.014	Benign	Cryptogenic cirrhosis	Abnormal IF assembly	[39]
c.448A>G	p.Ile150Val	K18	K8/18	Missense	0.247	Benign	Liver disease	Unknown	[40]
c.821T>C	p.Phe274Ser	K74	K74/14 or K74/28	Missense	0.998	Probably Damaging	Ectodermal Dysplasia, Pure Hair-Nail Type, ADWH	Disrupts 1B knob-pocket interaction	[16,41]

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
