# Peer review of "Molecular Modeling of Pathogenic Mutations in the Keratin 1B Domain"

_ijms, 2020, doi:10.3390/ijms21186641_

Round 1
Reviewer 1 Report
This work is interesting however one cannot escape the fact that it is an ”in silico” work and no experimental data from the wet lab is available in support of the conclusions. The data presented is novel, but I wonder how the authors see this work advancing the knowledge in this research area, as the structural (computer) model is a theoretical model above all.
In my opinion, the paper feels a bit ”flat”, as the final conclusions fall somewhat short of what the nice images/figures introduce us to. The discussion section to me lacks information in the sense how this extends the knowledge in the area.
Author Response
Response: We appreciate R1 recognizing the novelty of this work and the quality of the figures, and for the constructive comments about the in silico nature of the work, the biochemistry, and the knowledge in the field. In short, while this is in silico, all of this work is based off of our lab’s experimentally determined x-ray crystal structure of the K1/K10 1B tetramer. This is the only keratin tetramer ever experimentally determined, which is what put us in position to make high quality in silico observations. We agree that additional biochemistry should be done to examine and validate the properties of these mutations; but we believe this is beyond the scope of this manuscript which is about the modeling. We note that the COVID 19 pandemic, and its impact on our lab’s wetlab operations, precludes us from performing the biochemistry work at this time. To address these concerns, we have added the following:
An addition in the last paragraph of the Introduction (page 2 lines 66-67) emphasizing experimentally determined:
“The keratin structures represent the only heteromeric 1B structures to date, and as such they are quality experimentally determinedtemplates for modeling 1B domains from other keratin pairs.”
We also added a paragraph (second to last paragraph) in the Discussion on page 12 (lines 301-306):
“The molecular modeling presented here establishes a foundation for designing future biochemical studies to validate the predicted amino acid properties of these keratin mutations. While the modeling is in silico, this computational structural biology is based on an experimentally derived keratin 1/10 tetrameric crystal structure, and thus is more reliable than ab initiostructure prediction. The analysis of pathogenic K1 mutations mapped onto the 1B structure advances our knowledge of how disease-causing keratin mutations alter the various stages of mature filament assembly.”
Reviewer 2 Report
The manuscript entitled "Molecular modeling of pathogenic mutations in the
keratin 1B domain" describes a molecular modeling method describing the possible mutations in keratin.
the manuscript is well organized and discussed. The results were compared with literature and similar studies.
In my opinion the paper is suitable to be published in International Journal of Molecular Sciences after minor revisions.
Comments:
The abbreviations should be described when used for the first time
Figure should be written Figure and not Fig. in the main text
The references are not formatted according to the guideleines. Please correct.
Author Response
Reviewer 2: The abbreviations should be described when used for the first time.
Response: We have gone through the manuscript and updated all abbreviations with a proper description when used for the first time.
Reviewer 2: Figure should be written Figure and not Fig. in the main text.
Response: All instances of “Fig.” in the manuscript have been replaced with “Figure”.
Reviewer 2: The references are not formatted according to the guidelines. Please correct.
Response: The references have been updated to MDPI format.